# Is Perception Probabilistic?
Dobromir Rahnev, Ned Block, Janneke Jehee & Rachel Denison

## Scientific question
Is perception probabilistic? This is a grand question about the fundamental nature of how our mind operates and how this operation is implemented in brain circuits. The question is also interpreted differently by different people, making progress difficult. In this adversarial collaboration, four scientists with different backgrounds and perspectives on this issue[1–7] engaged in considerable discussion of how the issue of probabilistic perception can be operationalized as to be made empirically tractable. We have agreed that a critical dividing line between probabilistic and non-probabilistic views of perception is defined by the following question:

*Can humans perform flexible and deliberate (i.e., cognitive) computations that require access to more than summary statistics (mean and variance) of a perceptual likelihood function or posterior on a single-trial basis?*

## Background
Sensory input is noisy and ambiguous. For example, sensory noise makes it impossible for a tennis player to determine the exact landing location where a tennis ball hit the court[8]. Similarly, even if our senses were perfect, the world is inherently ambiguous such that the same 2D image on our retina is consistent with multiple interpretations of possible 3D objects in the real world. If achievable by the brain, the best possible way of dealing with such noise and ambiguity is to represent a full probability distribution over the possible world states, and several modern theories have postulated that our internal representations indeed operate in this fashion[9–18]. However, many other accounts tacitly assume that our perceptual system only represents a single guess that also features a sense of confidence in this guess[19–22]. So, which is it?

Empirical evidence has come mostly from two sources: studies on cue integration[23–25] and perceptual confidence[26,27]. More recently, several studies have measured sensory uncertainty directly in cortex and related it to behavior[4,5,28]. All of these lines of research clearly demonstrate that humans represent sensory uncertainty on a single-trial basis. Therefore, there is already extremely strong evidence against the view that the brain only represents a single point estimate (e.g., the mean of the probability distribution). Nevertheless, existing findings appear compatible with a slightly more complex – but still non-probabilistic – representation where only the mean and variance of the probability distribution are represented[2]. Thus, existing empirical evidence appears insufficient for establishing whether perception is probabilistic or not.

## Challenge or controversy
The challenge, then, is to determine whether sensory information is represented as a whole probability distribution or simply as a summary consisting of the distribution's first two moments (mean and variance). This question can be asked on many different levels. Here we focus on Marr's computational level[29] without regard for algorithmic or implementational considerations. Further, one can focus on the representations at different stages: (1) within the presumably automatic and potentially unconscious perceptual system, (2) at the interface between perception and cognition where perceptual representations can be used for flexible and deliberate computations (i.e., perceptual decision making), and (3) within cognition (for example, reasoning about possible world states). While there is likely to be controversy regarding all three stages, here we focus on the second stage: the

nature of perceptual information that is available for deliberate computations. We have chosen to focus on this stage and remain at Marr's computational level as we believe that this is a place of maximum controversy where progress can plausibly be made in a 1-to-3-year timeframe.

**Competing hypotheses and proposed approach for resolution**

The question that we focus on here is whether the single-trial perceptual information available for deliberate (though not necessarily conscious) computation is a full probability distribution or consists solely of summary statistics. This question can be tested by designing simple perceptual tasks that require access to more than summary statistics of a single-trial probability distribution.

- If humans can perform such tasks behaviorally, then this would strongly suggest that full probability distributions are constructed in perception and that these probability distributions are accessible for flexible and deliberate computations. This would thus be strong evidence FOR probabilistic perception.
- Alternatively, if humans cannot perform such tasks behaviorally even with plenty of training, this would strongly suggest that flexible and deliberate computations only have access to a summary of the population code. This would thus be strong evidence AGAINST probabilistic perception.

This question can be addressed behaviorally in multiple ways. One possibility is to use stimuli that produce a bimodal distribution on a single trial. Such stimuli can be constructed by mixing multiple directions of motion in a single stimulus[1,30] or by using a collection of individual stimuli drawn from a bimodal distribution in orientation or color space. The question would then be whether human subjects can perform a task that requires an accurate representation of both peaks of the distribution for briefly presented stimuli on a single trial. We suspect that many variants of such and similar tasks exist and are eager to receive further suggestions from the community.

**Concrete outcomes**

We believe that a series of behavioral experiments with converging findings (regardless of which hypothesis is supported) would have substantial impact on our understanding of the nature of perceptual representations. The strongest impact will of course be on theories defined on the computational level, but these results will meaningfully inform theories at the algorithmic and implementational levels as well.

**Benefit to the community**

Beyond addressing a question of fundamental importance for our understanding of sensory processing, this adversarial collaboration will have several additional benefits:

- It will pave the way for more inter-disciplinary collaborations. Our team members represent a synergy between cognitive neuroscience, computational neuroscience, philosophy, and psychology. These disciplines all independently investigate the same questions but with little cross-talk. The current effort can serve as a blueprint for future collaborations and further integration.
- It will further establish the relevance of behavior to neuroscience[31]. We believe that careful behavioral research can help constrain many computational and neural theories, and that this collaboration will lead to greater use of sophisticated behavioral paradigms in informing and even constraining theories of the neural implementation.

**Core group of committed collaborators**
AGAINST probabilistic perception

- *Dr. Doby Rahnev*, Assistant Professor of Psychology at Georgia Tech, USA. Rahnev has previously argued against full probability distributions in perception and has proposed alternative representational schemes. He will be involved in both the theoretical and empirical aspects of the proposal.
- *Dr. Ned Block*, Professor of Philosophy at NYU, USA. Block has recently criticized the evidence for probabilistic perception. He is a senior thought leader who will advise on whether the proposed experiments truly address the question at hand.

FOR probabilistic perception

- *Dr. Janneke Jehee*, Principal Investigator at the Donders Institute, the Netherlands. Jehee has previously demonstrated how sensory uncertainty can be decoded from population activity in the visual cortex and linked this uncertainty to behavior. She will be involved in both the theoretical and empirical aspects of the proposal.
- *Dr. Rachel Denison*, Assistant Professor at Boston University, USA. Denison has previously shown that humans incorporate different types of uncertainty in their perceptual decisions via behavior and modeling. She will be involved in both the theoretical and empirical aspects of the proposal.

**Statement of commitment**
We commit to collaborate on the chosen GAC topic, including:

- Incorporating feedback from the community and potentially welcoming new CCN community members to the GAC based on their written commentary to the GAC proposal
- Running an online kickoff workshop for CCN2020, inclusive of both founding core GAC members and those new members who joined through the community feedback process
- Writing the position paper to be submitted ~December 2020 to a curated special issue, to be accompanied by commentary pieces authored by attendees of the CCN2020 kickoff workshop
- Attending and presenting progress at the following CCN2021

Dobromir Rahnev, Ph.D.
Assistant Professor
Georgia Tech

Ned Block, Ph.D.
Professor
New York University

Janneke Jehee, PhD
Principal Investigator
Donders Institute

Rachel Denison, Ph.D.
Assistant Professor
Boston University

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
