# OpenReview forum: "Is Perception Probabilistic?"
_ccneuro.org/CCN/2020/Workshop/GAC_

### Official Review · ~Alex_Hernández-García1 · 2020-08-25
**Interesting and relevant question, clearly exposed. More details about experiments desirable.**

**Rating:** 7
**Soundness:** Strongly agree
**Confidence:** 4

**Review:**

The question of whether humans can perform computations that require access to more than summary statistics is very interesting and relevant to several scientific fields. I appreciate that this proposal emphasises the interdisciplinary nature of the question, in terms of the impact that resolving the question will make, and the collaborations that addressing the question requires. Furthermore, the goal of the proposal and the motivation are very clearly exposed.

While acknowledging that I have no specific expertise in the particular topic of the proposal, I have two concerns after reading the document that could be of help to further detail the future work derived from this proposal:
* I miss more specific examples of the kind of experiments that can be conducted to try to answer the question. The authors state that the question can be addressed in "multiple ways", but only one example is given: "stimuli that produce a bimodal distribution on a single trial". I think it will be beneficial for the subsequent studies, for better understanding of the proposal and for the transparency of subsequent studies to". I think it would help better understand the proposal and improve the transparency of subsequent studies to provide other examples and additional details about the possible experiments.
* I am not sure whether the question of whether the brain is capable of probabilistic perception is really binary, that is simply "against" or "for". For example, it may well be the case that the brain is able to perform computations that require access to more than summary statistics, but only within a limited family of probability distributions. It seems unlikely to me that the brain be able to fully process sensory data of arbitrarily complex probability distributions. If this was the case, the experiments should consider this option. For example, humans may not be able to perform a task involving stimuli that produces a bimodal distribution on a single trial (providing "strong evidence against probabilistic perception", according to the document), but they may be able to perform tasks involving unimodal or exponential distributions.

**Comments:**

As discussed before, the main strength of this proposal is that it addresses an interesting and relevant question for several communities. Furthermore, the document is written very clearly and the team is formed by authors with expertise in the topic and with diverse views (for and against the question).

As a weakness, I think the proposal could be more specific about the type of experiments that should be performed to answer the question. Finally, I am concerned that the answer to the question may not be as simple as yes or no, and therefore the more and more careful experimentation would be required.

**Controversy:**

Agree

**Definition:**

Agree

**Expertise:**

Strongly agree

**Outcomes:**

Strongly agree

---

> ### Public Comment · ~Doby_Rahnev1 · 2020-09-08
> **Author Reply**
>
> Dear Alex, thank you for your positive review of our proposal. We completely agree that our proposal lacks details about the behavioral experiments. This was actually intentional as our understanding is that the specific experiments should be the result of consensus in the community rather than designed in detail in advance. Our reading of the rest of the proposals is that they took similar approaches.
>
> Your second point about intermediate possibilities where more than summary statistics can be accessed but perhaps not arbitrarily complex distributions is an important one. In the end, what we are interested in (and believe that the community cares about) is not the labels we use but the reality of how the brain works. If the reality is more complex than the two positions outlined in the proposal, the hope is that the debate and resulting experiments will get us closer to it regardless of how the initial possibilities are constructed. That said, the two of us (Doby and Ned) who are skeptical of probabilistic perception would already need to substantially modify their positions if any clear example is found where perception represents more than a point estimate with heuristic uncertainty. And conversely, the other two of us (Janneke and Rachel) may need to modify their position if a series of experiments fails to find evidence for representations any more detailed than that. Therefore, while we do not claim that any one experiment will convincingly establish the nature of the perceptual representation, we think that progress can be made.

---

### Public Comment · ~Jean-Paul_G_Noel1 · 2020-08-11
**Bayesian decoder informed by efficient coding?**

Love the question, and one I think we as a field haven't valued enough. The panoply of evidence showing, for example, 'optimal cue combination' (e.g., Ernst & Banks, 2002) doesn't really demonstrate probabilistic coding. If I sample from one likelihood, then from another, and then do an an average of the two, I will -- overall, after many samples -- get the likelihood with expected mean and variance according to MLE, but the true question is indeed whether this is true on a single trial? If so, then the observer must have taken into account at least 2 moments.

But how about evidence from Bayesian decoders informed by efficient coding such as in Wei & Stocker 2015, Nature Neuroscience? There, they account for orientation estimations (which are repulsive, anti-Bayesian) by postulating likelihoods that are asymmetrical. Thus, for observers to produce these estimates, they must have known about the skewness of the likelihoods, no?

Can't wait to see what comes out of this great initiative!
jp

---

> ### Public Comment · ~Ulrik_R._Beierholm1 · 2020-08-12
> **Skewness**
>
> Jean-Paul, be aware that skewness of course depends on the parameter transformation, as exemplified by a log-normal distribution which has zero skewness in log-space. The brain may be able to do parameter transformations (e.g. log-transformation) easily, whereas bimodal or certain mixture distributions can not simply be parameter transformed. That said, it does certainly not rule out the use of measures of skewness in these type of studies, we just have to be careful how to interpret the results

---

> > ### Public Comment · ~Doby_Rahnev1 · 2020-09-08
> > **Author Reply**
> >
> > Dear Ulrik, thank you for explaining this issue. This is an important caveat about skewness and one of the reasons why we think that bimodal distributions may be the case where probabilistic and non-probabilistic accounts of perception are likely to differ the most.

---

> ### Public Comment · ~Doby_Rahnev1 · 2020-09-08
> **Author Reply**
>
> Dear JP, thank you for your comment. We agree that current cue combination studies are consistent with both probabilistic and nonprobabilistic interpretations. As for the Wei & Stocker (2015) paper, we think that it may provide evidence that people have access to more than mean and variance, but at present the evidence is just not strong enough. The biggest problem is that the results explained by their model (the repulsive biases) could be explained in other ways without postulating asymmetric likelihoods (e.g. Jazayeri, M. & Movshon, J.A. Nature 446, 912–915 (2007); Stocker, A.A. & Simoncelli, E.P. Adv. Neural Inf. Process. Syst. 20, 1409–1416 (2008)). In addition, as Ulrik mentioned below, in some circumstances it’s possible that a Gaussian internal representation corresponds to a skewed distribution of stimulus space, thus making skewness less diagnostic regarding the nature of the internal representation than, for example, bimodality. Nevertheless, we are open to the possibility that skewness could potentially be useful for distinguishing between probabilistic and non-probabilistic representation, if a suitable experimental design can be found.

---

### Public Comment · ~Ulrik_R._Beierholm1 · 2020-08-12
**Experimental tests**

I agree that there is certainly still an open question here, and the idea of testing if the computations are flexible, and uses more than simple summary statistics (i.e. Normal distribution) is a good starting point. These two aspects have usually been considered separate, i.e. testing can be done for flexible computations using different new priors/likelihoods (Bayesian transfer, 1,2), or using bimodal distributions (see 3,4 although not truly perceptual). Combining them is more interesting. Note that the use of mixture distributions could also be seen as a challenge to these ideas, if tested in a Bayesian transfer (1, 5, 6)

Looking forward to hear of future developments (and apologies for shamelessly plugging own papers)

1. Beierholm, Quartz & Shams
“Priors and likelihoods are encoded independently in a multisensory perceptual task”
Journal of Vision, 9: 23, 1-11 (2009).
2. Kiryakova, Aston, Beierholm, Nardini
“Bayesian transfer in a complex spatial localisation task”
Journal of Vision, 20(6):17 (2020)
3. Sanborn & Beierholm
“Fast and accurate learning when making discrete numerical estimates”
PLoS Comp Biology. 12(4): e1004859 (2016)
4. Spicer, Sanborn & Beierholm
“Using Occam's razor and Bayesian modelling to compare discrete and continuous representations in numerosity judgements”
Cognitive Psychology, 122, 101309 (2020)
5. Knill
Mixture models and the probabilistic structure of depth cues.
Vision Res, 43(7), 831–854 (2003)
6. Koerding, Beierholm, Ma, Quartz, Tenenbaum & Shams
“Causal inference in multisensory perception”
PLoS One 9, e943 (2007).

---

> ### Public Comment · ~Doby_Rahnev1 · 2020-09-08
> **Author Reply**
>
> Dear Ulrik, we are glad that you seem to agree with the basic premise of our proposal. And thank you for the very relevant references! These papers provide clear evidence that the brain can represent binomial distributions and can have complex priors. However, as you point out, this is not necessarily the same as having probabilistic perception because these findings are also consistent with cognitive representation of probabilities coupled with non-probabilistic perception. This is why we think that it’s critical to test whether bimodal distributions can be represented by the sensory system in conditions where complex priors and the role of cognition are minimized. Will be curious to know if you have ideas about specific experimental designs that can accomplish this.

---

### Public Comment · ~Xaq_Pitkow1 · 2020-08-14
**Why do you need more than a normal distribution to do probabilistic computation?**

The essence of probabilistic computation is using uncertainty or confidence, as opposed to a point estimate. But this GAC proposes you need more than just a single measure of uncertainty to conclude that computation is probabilistic. That would certainly provide a richer basis for probabilistic computation, and I believe it exists, but it's unclear why that is necessary.

Another key question is, what is the uncertainty about? And if the identity of that latent variable is itself unknown, then transformations of a normal latent variable can create all sorts of other distributions (including multimodal distributions, like tanh(x) for x~normal).

---

> ### Public Comment · ~Doby_Rahnev1 · 2020-09-08
> **Author Reply**
>
> Dear Xaq, thank you for your comment. There are three points in your comment that we address below: (1) the definition of probabilistic perception, (2) the distinction between probabilistic perception vs. probabilistic computation, and (3) transformations of variables.
>
> 1. The definition of probabilistic perception
> As far as we are aware, there is no firmly agreed upon definition of what probabilistic perception refers to exactly. Thus, one of the contributions of this GAC should perhaps be to establish such an agreed-upon definition in collaboration with the community. It is true that if probabilistic perception is defined as anything more than a point estimate, then the existence of probabilistic perception is completely uncontroversial: all 4 of us agree that perception gives you more than a point estimate and we believe that virtually everyone in the community agrees too. This consensus is great but it also obscures important differences that exist. For example, one of the main alternatives to established theories like Probabilistic population codes (PPC), Neural sampling, and Distributed distributional codes (DDC) is that the perceptual representation consists of a best guess (i.e., a point estimate) plus heuristic uncertainty that may only partially reflect the true uncertainty. It appears to us that such a representation should not be called “probabilistic” as probabilities can only be extracted from it in special cases of certain (e.g., Gaussian) distributions and by making additional assumptions regarding the brain’s ability to compute the probabilities associated with any point on the axis given a mean and a variance. Nevertheless, the terminology is not critical here. If the community feels that this type of representation should be called probabilistic, then perhaps we can come up with a different term to designate the dividing line between representations based on a point estimate and heuristic uncertainty vs. representations that consist of more detailed information as afforded by PPC, DDC, and neural sampling. In response to this and several other reviews, we plan to include a focused discussion on the term “probabilistic”  and are prepared to change the terminology if necessary.
>
> 2. The distinction between probabilistic perception vs. probabilistic computation
> Your comment appears to be exclusively about probabilistic *computation*. We all think that the notion that the brain can perform at least some versions of probabilistic *computation* is virtually an established fact. For example, we can reason about the probabilities associated with rolling dice or playing card games. The point of contention is rather whether a representation of probabilities (more detailed than a point estimate coupled with heuristic uncertainty) is present in the output of the sensory system and used in downstream computations, or not.
>
> 3. Transformations of variables
> This point has come up in a couple of comments. It is true that some transformations of unimodal variables result in multimodal variables. This means that even if the true representation is of only a point estimate coupled with heuristic uncertainty, then in some cases a multimodal distribution can be extracted. However, this multimodal distribution will have the same shape on every trial and it would not be a good representation of, for example, a bimodal distribution where the two modes can be experimentally varied from trial to trial. Therefore, we think that experimental designs where the two modes of a bimodal distribution can vary across trials can distinguish between simple representations like point estimate with heuristic uncertainty and more complex ones like what is afforded by PPC, DDC, and neural sampling.

---

### Public Comment · ~Tyler_BrookeWilson1 · 2020-08-24
**Controlling for Mode Collapse**

I’m excited about this proposal and the experiments to come! I had one concern about how you characterized the hypothesis space though. You write that you’re interested in whether the outputs of perception (or vision) include full distributions or merely partial summary statistics. I’m worried that these two hypotheses aren’t exhaustive, and that remaining option might make trouble for the interpretation of your results. The other option in this space is that perception computes posteriors by sampling and that the outputs of perception are the approximate posterior given by the full set of samples (Gershman et al. 2012, Vul et al. 2014). This could be a small number of samples if the proposal distribution is good (Kulkarni et al. 2015).

The reason this is relevant to an experiment aimed at the Marr computational level, rather than the algorithmic level, is because the sampling hypothesis makes the prediction that how much of a distribution is represented will differ depending on other features of the experimental set up. For example, if the sampling hypothesis is right, there may be a moment in processing (perhaps very brief) when the subject has only drawn a single sample, and so will seem to represent only a very noisy mean of the distribution, and no variance. A few moments later the subject may have drawn only a small number of local samples, and will so may only represent one mode of a multi-modal distribution (and may seem to represent a noisy mean and variance). Given enough processing time however, and provided samples are not internally discarded, the subject will come to represent multiple modes (and so the full distribution, at least to a rough approximation).

The problem is not just with time. Other aspects of the set up could also lead to such ‘mode collapse’, such as if the stimuli are such that one mode of the posterior is much higher probability than the other(s). The worry for your experiments is that mode collapse might lead to a false negative result, if subjects don’t have enough time or if the differences between modes are too great. If you controlled for these, I’d be more prepared to accept a negative result (which I would interpret as the result that people only represent some degenerate approximation of the full posterior, either mean and variance or small N approximations). Alternatively, if you found that participants had access to full, multi-modal distributions late in processing, but not early, that would be strong evidence for the sampling hypothesis (an algorithmic level insight as a bonus on a computational level experiment!)

Gershman, S. J., Vul, E., & Tenenbaum, J. B. (2012). Multistability and perceptual inference. Neural Computation, 24(1), 1–24. https://doi.org/10.1162/NECO_a_00226
Kulkarni, T. D., Kohli, P., Tenenbaum, J. B., & Mansinghka, V. (2015). Picture: A probabilistic programming language for scene perception. Proceedings of the IEEE Computer Society Conference on Computer Vision and Pattern Recognition, 07-12-June, 4390–4399. https://doi.org/10.1109/CVPR.2015.7299068
Vul, E., Goodman, N., Griffiths, T. L., & Tenenbaum, J. B. (2014). One and done? Optimal decisions from very few samples. Cognitive Science, 38(4), 599–637. https://doi.org/10.1111/cogs.12101

---

> ### Public Comment · ~Doby_Rahnev1 · 2020-09-08
> **Author Reply**
>
> Dear Tyler, we completely agree with you - certain behavioral results may seem non-probabilistic on the surface but be completely consistent with neural sampling. We are fully aware of this issue and will take it into account in the design of experiments. It is also arguably unlikely that a single behavioral experiment will convincingly distinguish between probabilistic and non-probabilistic accounts of perception, and thus this is not necessarily the goal of our proposal. Rather, we want, in consultation with the community, to establish what kind of behavioral evidence would be relevant to this debate and conduct the behavioral experiments that seem the most promising. As most things in science, it is likely that the results of these early experiments will inform the debate but not resolve it. It is also likely that beyond simple qualitative results, we would need to fit computational models based on, for example, probabilistic population codes and neural sampling directly to the data. This approach can ensure that a mode collapse hasn’t occurred. Overall, we agree with your comments fully and realize the challenges ahead. That said, we believe (and it seems that you also agree) that these challenges are not unsurmountable as long as they are carefully taken into account.

---

### Public Comment · ~Reny_Baykova1 · 2020-08-24
**Representations of variance**

This is a very interesting proposal. One of the main questions I am investigating in my PhD is whether observers track specifically the variability of sensory information presented over time. The existing evidence for this is limited because it is confounded by changes in other distributional statistics (e.g. range, skewness). Very broadly speaking, if perception is probabilistic, observers’ responses on any given trial will be biased towards the stimuli that are most likely to appear throughout the course of the entire experiment, and this bias should be proportional to the variance of the underlying stimulus distribution – the smaller the variance, the greater the bias. We are currently investigating this prediction in two experiment within the domain of duration estimation. You can read our stage 1 registered report (Baykova, Buckley, Seth, & Roseboom, 2019) here: osf.io/6phx7.

---

> ### Public Comment · ~Doby_Rahnev1 · 2020-09-08
> **Author Reply**
>
> Dear Reny, thank you for referring us to this work, which looks very interesting. That said, it seems that you’re asking a slightly different question than what we’re investigating here. It appears that your project is investigating whether people can form a prior from exposure and then use it according to the principles of Bayesian computation. We think that this question is very interesting (all four of us would predict that the answer is yes) but this could occur even if the likelihood function on each trial is non-probabilistic (which is the point of contention here). We couldn’t emphasize this point much in our proposal but we plan to expand on it much more if our proposal is invited to move on: the fact that people can perform computations in accordance with Bayes’ theorem does not directly imply that representation of sensory evidence on a single trial is probabilistic or not.

---

### Public Comment · ~Raphael_T_Gerraty1 · 2020-08-25
**An abnormal definition of probabilistic representation?**

This proposal is an attempt to answer the question of whether our perceptual system uses probabilistic representations. This is an important area of research and one in which clarification of what makes a representation probabilistic is sorely needed. However, it is not clear how to justify the necessary condition – that a probabilistic representation must be something beyond a normal distribution – which grounds the framework presented here. This limits the conclusions we will be able to draw from the proposed collaboration.

Much of the trouble stems from the claim that evidence so far supports only “a slightly more complex – but still non-probabilistic – representation where only the mean and variance of the probability distribution are represented”. For some generative models, the optimal posterior distribution is gaussian. For some probabilistic methods like variational bayes, a gaussian distribution might be used to approximate the optimal posterior as closely as possible. A gaussian distribution is fully specified by a mean and variance. In what sense could a probability distribution for which mean and variance are sufficient statistics be called “non-probabilistic”? The authors seem to have a distinction in mind between the representation of a probability distribution and a probabilistic representation, perhaps based on some notion of the distribution’s complexity or its sub-optimality when conditioned on a specific generative model, but such a distinction is not spelled out or justified.

Even if we knew that a particular posterior distribution should be non-gaussian (i.e. because we knew the prior and likelihood of the brain’s generative model governing some inference), there is no obvious reason to exclude a gaussian approximation to that posterior from the definition of probabilistic representation, and none is provided. In addition, any parametric probability distribution can be fully described by sufficient statistics. Unless we are willing to exclude, say, exponential family distributions as a whole, stating that mean and variance are not probabilistic, while other summary statistics are, seems arbitrary without further clarification. Would a noncentral t-distribution count as a probabilistic representation if degrees of freedom were represented? What about a mixture of gaussians? It is not clear from the framework described here.

Because of these ambiguities, it does not seem helpful to us to define probabilistic representations based on whether summary statics or “full probability distributions” are being represented. It is clear that we have to draw the line someplace: point estimates could be interpreted as specifying delta distributions, but in this case probability would not be a helpful or parsimonious label. So what do we mean by probabilistic representation? When is it useful to invoke probability?

The essence of probabilistic representation should be the use of probability distributions to represent uncertainty. What is needed for a representation to be probabilistic is that uncertainty about a variable in an internal computation be expressed using a probability distribution in a way that approximates conditional probabilistic (bayesian) inference for the generative model assumed. Seen through this lens, whether the probability distribution is represented via something like MCMC sampling or the use of sufficient statistics does not matter for the general question of whether a neural computation involves probabilistic representation.

The important question would instead be whether and how the representation of uncertainty guides the behavior of a system. What is needed are experiments that can distinguish between, for example, tracking the mode of a distribution vs marginalizing over it. This would actually be one reason to avoid invoking mean-and-variance-based posteriors in simple experimental contexts with i.i.d. observations, but as the authors have pointed out here, experiments can be cleverly designed to model the use of gaussian distributions to represent uncertainty in cue combination and other tasks, which seem to rule out representing only the mode.

None of this means that the question of whether there are limits on the expressiveness of algorithms our brains use to represent uncertainty is not interesting. Experiments showing that we could not represent perceptual uncertainty with a bimodal distribution, for instance, would be an important contribution to our understanding of the perceptual system being measured. But, they would not necessarily help us answer the question “Is perception probabilistic?”.

--Raphael T Gerraty and Samuel Lippl

---

> ### Public Comment · ~Doby_Rahnev1 · 2020-09-08
> **Author Reply**
>
> Dear Raphael and Samuel, thank you for the detailed response. We mostly agree with you but there are a few points of our original proposal that we want to clarify.
>
> The main alternative to probabilistic perception that we have in mind is that the perceptual representation always consists of a best guess (i.e., a point estimate) coupled with a potentially heuristic notion of uncertainty. It appears to us that such a representation should not be called “probabilistic” though if there is a disagreement about this terminology (e.g., Xaq Pitkow appeared to disagree in a previous comment), then different terminology could easily be adopted. (By the way, we really like the definition you provide: “The essence of probabilistic representation should be the use of probability distributions to represent uncertainty. What is needed for a representation to be probabilistic is that uncertainty about a variable in an internal computation be expressed using a probability distribution in a way that approximates conditional probabilistic (bayesian) inference for the generative model assumed.” It seems to us that a point estimate with heuristic uncertainty would not be counted as probabilistic representation under this definition for cases where the underlying distribution is bimodal.)
>
> Further, we want to clarify that our claim is not that a Gaussian representation should not be counted as probabilistic when that is the correct distribution to represent. Instead, what we want to argue is that when the underlying likelihood is Gaussian, probabilistic and non-probabilistic schemes can make the exact same predictions. This is why we think that we need to use other distributions (preferably multimodal) to distinguish between probabilistic and non-probabilistic accounts. It is definitely not our intention to arbitrarily label Gaussian distributions as non-probabilistic.
>
> In the end, we think that behavioral experiments can make progress on the issue of whether multimodal distributions can be represented in perception (which seems to be implied by current accounts that describe perception as probabilistic like PPC and neural sampling). If the community doesn’t think that “probabilistic perception” is the best way of describing this endeavor, we can change our terminology. But in parallel with this, we need to decide as a field what counts as “probabilistic perception” and how this can be tested experimentally. We hope that this proposal will result in a robust and productive discussion on this issue that will lead to such an agreed-upon definition.

---

### Public Comment · ~Krzysztof_Basiński1 · 2020-08-26
**Probabilistic perception or parameter representations?**

This is a very interesting proposal addressing an important topic which may improve cross-talk between philosophy, cognitive science and neuroscience. However, in line with what Raphael Gerraty and Samuel Lippl pointed out before, I am not sure about the use of the term “probabilistic perception”. As gaussian probability distributions may be fully described using mean and standard deviation (summary statistics), should a representation based only on mean and standard deviation also count as probabilistic perception? On the other hand, multimodal distributions can be approximated using sets of means, standard deviations and mixing parameters. These may also be thought of as “summary statistics”.  Perhaps the central question should be rephrased to stress WHAT is represented in the brain - actual distributions or some parameters of these distributions?  I’m not sure however if this can be answered without looking at the implementation / neurons doing the number crunching.  Or, judging from the proposed experiments, perhaps the question should stress the unimodality vs. multimodality of distributions?

I would also like to make a suggestion about the design of the planned experiments. The proposal mentions experiments in the visual domain, but I would suggest looking at other senses as well. For example, it is reasonable to assume that audition works quite differently than vision and that auditory features (such as pitch or timbre) are represented differently than visual features (Griffiths, 2012; McDermott, 2018).  I think that there is a need to design similar experimental tasks in multiple sensory modalities if the results are to be generalised to all of perception and not just to vision. One possible paradigm that can be of use involves manipulating harmonicity in pitch perception tasks (McPherson & McDermott, 2018). People have a salient pitch sensation if multiple frequencies in the sound form a harmonic series. Distorting the series may lead to inability to perform pitch comparisons. Distortions can be manipulated in various ways, for example forming a bimodal distribution.


Griffiths, T. D. (2012). Cortical mechanisms for pitch representation. *Journal of Neuroscience*, *32*(39), 13333–13334. https://doi.org/10.1523/JNEUROSCI.1661-12.2012

McDermott, J. (2018). Audition. In J. T. Wixted (Ed.), Stevens Handbook of Experimental Psychology and Cognitive Neuroscience, Fourth Edition (pp. 1–56). Wiley.

McPherson, M. J., & McDermott, J. H. (2018). Diversity in pitch perception revealed by task dependence. *Nature Human Behaviour*, *2*(1), 52–66. https://doi.org/10.1038/s41562-017-0261-8

---

> ### Public Comment · ~Doby_Rahnev1 · 2020-09-08
> **Author Reply**
>
> Dear Krzysztof, thank you for your comment and suggestions for experiments. As you mention, the issue of what counts as probabilistic perception has come up a few times in previous comments. We think that a meaningful distinction can be drawn between models that assume that the perceptual representation consists of only a point estimate coupled with heuristic uncertainty, and more detailed representations that include full probability distributions (like PPC and neural sampling). We further think that this distinction corresponds well to the concept of “probabilistic perception” where the former representations are non-probabilistic but the latter are. However, as we mentioned below, if there is disagreement about terminology, we are prepared to adopt different terminology. That said, we would still need to address as a field what exactly is meant by “probabilistic perception” and how this can be tested. We hope that our proposal will help in this effort. Nevertheless, we agree that the reality of what is going on is much more important than the label that we use to describe it and this is ultimately what we are trying to discover in this proposal. Finally, our focus on vision is driven by the expertise of the four authors who all work mostly in that domain. Your proposal about auditory experiments sounds really interesting. If our proposal moves forward, we would be thrilled to have a more detailed discussion about how the same question can be tested by varying the pitch of auditory stimuli.

---

### Public Comment · ~Ana_Todorovic1 · 2020-10-02
**Important to separate stimulus likelihood from base rate occurrence**

This is a very interesting question and I look forward to seeing the outcome of this adversarial collaboration.

I wanted to point out that it is normally quite difficult to separate out effects of adaptation/fatigue from effects of likelihood, because likelihood is often manipulated by an increase in base rate. Interestingly, Horace Barlow had a model of neural adaptation that also rested on capturing mean and variance:

	Barlow, H. B. (1961). Possible principles underlying the transformation of sensory messages. Sensory communication, 1, 217-234.

I’m not sure if conflation with adaptation can be avoided when presenting bivariate distributions. Usually conditional probability designs are necessary in order to capture only probability of stimulus occurrence while holding base rate equal across stimulus identities. (This is achieved either by presenting probabilistic cues in another modality or by presenting stimuli in one modality sequentially using a Markov chain: e.g. A predicts B more often than C, B predicts C more often than A, C predicts A more often than B.)

Another question that comes to mind in the scenario of a bimodal distribution is whether the mean is represented as well even if it is never displayed. Might be two peaks of the bimodal distribution be represented as distance-from-the-mean? It would be important for the models to account for this possibility.

---

### Public Comment · ~Tsvi_Achler1 · 2020-10-15
**There are more than two options**

I really like this topic and format.  It touches some of the core issues that has been plaguing computational neuroscience form the beginning.

The main question is: “is the single-trial perceptual information available for computation a full probability distribution or consists solely of summary statistics?”  I think this is an oversimplified question unless the question is referring to popular models and asking whether solutions are more likely achieved using feedforward neural network mechanisms (note even recurrent neural networks use a feedforward mechanism) through fixed weights summarizing feedforward learning statistics, or solutions are achieved using an explicit Bayesian-like calculation with priors, and distributions.

In that case I think neither model is sufficient.  Bayesian solutions are not scalable but are the most brain-like because priors can be included.  However Neural Networks solutions are not practical either because the rehearsal required to get the summary statistics introduces inflexibility to apply priors and quick updates.

But there are other possibilities.  The brain may learn simple average-like likelihoods as weights and then estimate distributions during recognition when the recognition context is present, allowing the inclusion of priors.  A connectionist a model implementing this approach requires optimization during recognition, not to find weights but to find activations.  And its simple likelihood learning is very fast so can include more features.  This is an new idea that I find very difficult to introduce when the scope narrowly looks at problems from basically only two types of models https://youtu.be/QP_3gGT90nE .

Reference to published model: Achler T, Symbolic neural networks for cognitive capacities, Biologically Inspired Cognitive Architectures, Volume 9, July 2014, Pages 71-81 https://doi.org/10.1016/j.bica.2014.07.001

I wish there would be more openness to consider other options as I personally find progress is inhibited immensely.